


# Towards systematic planning of small-scale hydrological
# intervention-based research
**Kharis Erasta Reza Pramana[1] and Maurits Willem Ertsen[1]**
[1]{Water Resources Section, Faculty of Civil Engineering and Geosciences, Delft University
of Technology, The Netherlands}
Correspondence to: K.E.R. Pramana (k.e.r.pramana@gmail.com)
**Abstract**
Many small-scale water development initiatives are accompanied by hydrological research to
study either the shape of the intervention or its impacts. Humans influence both, and thus one
needs to take human agency into account. This paper focuses on the effects of human actions
in the intervention and its associated hydrological research, as these effects have not yet been
discussed explicitly in a systematic way. In this paper, we propose a systematic planning,
based on evaluating three hydrological research projects in small-scale water intervention
projects in Vietnam, Kenya, and Indonesia. The main purpose of the three projects was to
understand the functioning of interventions in their hydrological contexts. Aiming for better
decision-making on hydrological research in small-scale water intervention projects, we
propose two analysis steps, including (1) possible surprises and possible actions and (2) cost-
benefit analysis. By performing the two analyses continuously throughout a small-scale
hydrological intervention-based project, effective hydrological research can be achieved.
**1 Introduction**
Small-scale water development initiatives play an important role in supporting sustainable
water resources management. Such projects are usually initiated and/or supported by local
non-governmental groups, but also by larger donors such as USAID and others (Van Koppen,
2009; ECSP, 2006; Warner and Abate, 2005). Typical small-scale intervention projects
include water harvesting development, improving small-scale irrigation schemes, and small
dams for water use or hydropower (Lasage et al. 2008; Ertsen et al., 2005; Falkenmark et al.,





2001; Farrington et al., 1999). A basic understanding of the local hydrology is typically
required for design, construction and management of small-scale water interventions. Even
though such a hydrological study may be limited in scope – both in terms of time and detail –
it still takes considerable effort performing the study and collecting the data. This holds
especially for building and maintaining (informal) networks and relationships for successful
local data collection (Mackenzie, 2012).
Many small-scale water intervention projects, especially those in the so-called developing
countries are located in areas that have been studied less well. In 2003, the International
Association of Hydrological Sciences (IAHS) initiated the Prediction in Ungauged Basins
(PUB) initiative with the objective to promote the development and use of improved
predictive approaches for a coherent understanding of the hydrological response of ungauged
and poorly gauged basins (Sivapalan et al., 2003; Hrachowitz et al., 2013). Our hydrological
studies in remote areas in Vietnam, Kenya and Indonesia were originally in ungauged
catchments as well. Our approach was based on investigating dominant hydrological
processes through a multi-method approach (Mul et al., 2009; Hrachowitz et al., 2011). Our
studies were in short field campaigns within strict financial constraints (compare with Mul et
al., 2009; Hrachowitz et al., 2011). On-site measurements were highly dependent on the
support of the local communities.
Small development activities have been well studied. Phalla and Paradis (2011), Gomani et al.
(2009), and Das et al. (2000) discuss hydrological research and local participation in
interventions to improve decision-making for interventions. In order to implement properly an
intervention, theories and practices of adaptive management have been suggested as potential
beneficial approaches (Fabricius and Cundill, 2014; Beratan, 2014, Von Korff et al., 2012).
Furthermore, local participatory approaches in hydrological monitoring throughout the world
have shown to be potentially effective – e.g. in South Africa, Zimbabwe, and India (Kongo et
al., 2010; Vincent, 2003; Das 2003; Das et al, 2000). However, combined focus on both
hydrological research design/management and local participation in hydrological research has
been rather absent from the literature. Currently, a more systematic overview of issues on
planning hydrological research within small-scale water intervention projects is lacking. This
paper aims to fill this gap.
We start with a discussion on human agency within hydrological research, including issues of
participation. Then, we discuss our experiences in our three case studies. These summaries are





meant to allow the discussion on scenario development in the third part of this paper where
we will discuss the social realities of the projects. We will contextualize the realities with our
chosen participation theory and approaches, which will be discussed at a basic level. Finally,
we propose how to plan hydrological research in a (surprisingly) surprise-rich context in a
systematic way.
## 2    Human agency and hydrological research
When we started our projects, not all that were to happen was, or could be foreseen. This
paper is an attempt to make sense of the events afterwards. In doing so, we traced the social
processes relevant for the development of research and intervention in our three cases and
looked for patterns. In the current context, as hydrologists who cannot be separated from the
socio-hydrological world (Lane, 2014), we searched for a better way of conducting small-
scale hydrological research in the future. How can hydrologists make better decisions when
planning hydrological research realizing that humans make decisions on a daily basis that will
affect the intervention development and hydrological research itself? Our objective is to
propose a systematic process of performing hydrological research in small-scale water
intervention projects. We argue that more explicit attention helps to design more appropriate
answers to the challenges faced in field studies. In particular, we propose two related steps:
(1) take into account possible surprises and resulting actions, and (2) using cost-benefit
analysis to analyse the need for certain measurements and assess effects of human
intervention.
Humans change landscapes through interventions for many purposes due to human demands
(Ehret et al., 2014). Hence, human agency is continuously changing future hydrology, which
means we need to build deeper understanding of human-water dynamics (Sivapalan et al.,
2014; Ertsen et al., 2014). In our cases, it turns out to be highly relevant to look at the
interactions between humans (as initiator and/or stakeholder of intervention and/or research
itself) and the complex hydrological system. Likewise, as the interventions influence society -
beneficially or not – societal actors (need to) create an awareness and overall understanding of
the interventions. Hydrological change usually occurs after a certain intervention has been
implemented. On the other hand, societal actors actually interact before, during, and after the
intervention. Therefore, potential interactions with possible feedbacks and changes not only
show that humans play an important role in determining much of the behaviour of





catchments, but also may already influence hydrology - and consequently society - before the
original hydrological effects of an intervention would have shown themselves.
Many studies of small hydrological research related to interventions – if available at all –
include human agency in the research through the lens of theft and vandalism (see Kongo et
al., 2010; Mul, 2009; Gomani et al., 2009). When theft and vandalism enter the debate, they
seem to be perceived as simple bad luck, which could happen every time and everywhere
during a research effort. This may be true in itself, but one should be aware that human
interventions in hydrological studies are not always similar to theft/vandalism. Perhaps people
interfere with measuring equipment out of curiosity, or because they simply do not know
what it is. There might be cases when certain agents are against the measurements being taken
in the first place, or are against measurements at a certain location – as will be shown below
when we discuss how motivations of stakeholders to interfere in one hydrological campaign
changed over time, without theft ever being a motivation for action. Whatever the case,
human intervention usually results in lower data availability. Especially as data sets would
have been relatively limited anyway, studies using such limited data are even more difficult to
be accepted in the scientific research community (compare with Winsemius, 2009).
In order to be able to design responses during hydrological studies, we argue that human
agency – both positive and negative – should be an integral aspect of designing, performing,
and evaluating intervention-based hydrological research. In terms of planning for surprises,
we have found the frameworks, as developed by the RAND cooperation on how to be
prepared when facing "surprises" in planning, extremely useful. Dewar (2002) (see also
Dewar et al., 1993) discusses such surprises and provides a tool for improving the adaptability
and robustness of existing plans by making assumption-based planning (ABP). With ABP,
one would double-check the planners' awareness of uncertainties associated to any plan,
including assumptions that might have been overlooked. In addition, research budgets for
small-scale interventions are usually constrained (e.g. Phalla and Paradis, 2011). What to do
with such limited budget, how human action affects research activities and budget, and how to
deal with possibly costly surprises are important questions to prepare oneself for. In terms of
time constraints, a very useful example of how to optimize short-term data is offered by
Hagen and Evju (2013). To understand a certain water intervention, ideally a hydrological
researcher would prefer measurements being conducted at many locations, for a long time and
with high frequency. However, within that general preference and given financial constraints,





much remains to be chosen by the researcher (Hamilton, 2007; Soulsby et al., 2008). This
suggests that different researchers would select different actions and measurement techniques,
even when performing a similar type of hydrological research. As such, choices can be
studied in terms of costs and benefits.
Despite this potential of looking at uncertainty in planning of small-scale hydrological
research related to human actions, there is still a long way to go. The above-mentioned bias
towards not publishing small-scale studies not only may limit understanding of the hydrology
of small-scale water systems, but it also prevents understanding the nature and performance of
the small-scale studies in relation to the intervention itself. Any intervention can be
understood in terms of cooperation and negotiation between actors in the process of
(re)shaping its design (Ertsen and Hut, 2009). In other words, water planning and
management are typically organised or 'co-engineered' by several agencies or actors (Daniell
et al., 2010). This co-engineering will also be the case in shaping the hydrological research
itself – and thus principally the science of hydrology as well.
In this paper, we evaluate co-engineering of the hydrological sciences in action. We scan for
solutions, explicitly analyse research management in the three cases, and define how it can be
improved in practice (see Sutherland, 2014). Daily realities of performing small hydrological
studies are our focus. Based on evidence of the effectiveness of our own learning, we
contextualize our personal experiences to extrapolate towards general principles how to
improve knowledge development for researchers and practitioners (Beratan, 2014). Below,
we first discuss the empirical findings from our three field studies.
**3    Three small hydrological research projects**
**3.1    Vietnam Case: Contour trenches for artificial recharge in Ninh Thuan**
**Province**
Contour trenching is one of the water harvesting techniques implemented to increase water
availability in semi-arid and arid region. A study on trenches in Chile by Verbist et al. (2009)
suggested that few efforts were observed to quantify the positive effect of runoff water
harvesting techniques on water retention. On the other hand, Doty (1972) found that there is
almost no change in soil water between areas with and without trenches. In this study, we
investigated recharge processes of contour trenching.





The study area is located in the Phuoc Nam Commune, in the Ninh Phuoc district, with
latitude 11º 27' 46.06" and longitude 108º 55' 44.39" E (Fig. 1). The landscape is a foothill
with an average slope of about 3.5%. It is dominated by mountainous granite and downhill
valley with mix of loamy sand, weathered granite, residual soils and alluvial deposits. The
climate is dominated by tropical monsoons. Generally, the wet season with heavy rainfall
events occurs from September to December. However, from April to May there are
sometimes light rainfall events. The dry seasons are from January to April and from June to
August. The average rainfall is 810 mm year$^{-1}$. Much of the area could be seen as bare soil
with erosion gullies, with some parts covered by cacti and grasses. Initially, contour trenching
of 4 m wide and 1 m deep was planned for an area of 97 ha. At the end of the project, only 22
ha were trenched with a combination of 4 m wide, 1 m deep and 1 m wide, 0.8 m deep
trenches. The research area itself focused on an area of about 8 ha.
We conducted a multi-method approach during a single wet year in 2009. Only during one 6-
month period (June to November 2009) data of rainfall, water levels and groundwater levels
were simultaneously available. On 11 October 2007, before the construction of contour
trenches, we installed two rain gauges (Casella tipping buckets, 0.2 mm, with HOBO data
logger). Infiltrated ponding water in the trenches was monitored daily after rainfall events.
The surface water was measured using a stick and two measuring scales in the trenches. The
subsurface (geology and soil) survey was partially conducted three times, in October 2007,
March 2008, and June 2009. In October 2007, we performed inverse auger tests (Porchet and
Laferrere, 1935) to obtain a range of hydraulic conductivity values. Six locations were
selected from uphill to downhill in the proposed trench area. In March 2008, we drilled 3 m
holes at eight locations for additional lithology investigations. We also took four soil samples
at the trench plots and analyzed those using wet and dry methods at the laboratory of
UNESCO-IHE Delft. In October 2007, we constructed three observation wells; Wells 1, 2,
and 3, at the upstream and middle part of the planned project area. We measured the
groundwater level with divers (Schlumberger Water Services, Delft, The Netherlands).
Because of the loss of two divers in two different observation wells, since February 2008 we
measured groundwater level manually on a 3 to 4-day basis. Moreover, we performed an
isotope study; as such studies on recharge are known to allow improved understanding of
catchment dynamics (Soulsby et al., 2003; Rodgers et al., 2005; McGuire and McDonnell,
2007). From September 2009 to November 2009, 72 water samples were collected in 2 ml
vials. $^{18}$O variation of rainfall, surface water and groundwater were analysed at the Isotope



Laboratory of Delft University of Technology. Field measurements and isotope results were
then used in Hydrus (2D/3D) modelling (Šimůnek et al., 2008).
We conclude that the combination of field measurements, the isotope technique, and
modelling over a 6-month period has given us the understanding of the recharge process in
contour trenching plots in Vietnam. Based on the groundwater level measurements and
obtained isotope signature in the groundwater, artificial recharge took place in the trench area.
From the modelling in Hydrus (2D/3D), the estimated values of parameters used were focused
on matching the true scenarios of possible hydraulic conductivities and porosities. Moreover,
in the long term, infiltration in the trenches will increase the groundwater levels based on the
events during the wet season. The quick groundwater level increase is followed by gradual
drawdown during the dry season. For the time being, the trenches seem to benefit short-term
subsurface storage.

### 13   3.2   Kenya Case: The impacts of contour trenches in Amboseli, Kenya

In Amboseli, a semi-arid area in Kenya, contour trenching started in 2002. Until recently, the
hydrological long-term impacts of this construction were not well documented. Previous
studies showed impacts of similar water harvesting techniques in different dimensions and
semi-arid areas e.g. Makurira et al., (2010), Singh (2012), Mhizha and Ndiritu (2013). An
attempt was made to answer two research questions on the impacts of contour trenching. First,
what is the impact of trenching on vegetation growth? Second, what is the impact of trenching
on soil redistribution in the trench area?
The contour trenching area is located about 30 km downstream of Kilimanjaro Mountain (Fig.
2). It lies at the altitude of 1,245 m, with latitude 2° 46' 57.46" S and longitude 37° 16'
45.93"E. The topsoil is sandy clay with volcanic rock found in deeper layers. The average
rainfall is about 400 mm year$^{-1}$. From visual observation, the study area was eroded and has
an average slope of about 2%. Additionally, it is situated next to an erosion gully, which
originated from Kilimanjaro Mountain. There were two types of trenches; first 1 m wide, 0.8
m deep and second 4 m wide, 1 m deep. From 2002 until recently, a temporary diversion
structure from stones was made to divert upstream rainwater to the whole trenched area.
For vegetation growth analysis, two types of satellite images were used; Tropical Rainfall
Measuring Mission (TRMM) and Moderate Resolution Imaging Spectroradiometer (MODIS)
- Normalized Difference of Vegetation Index (NDVI) time series were downloaded freely





from https://wist.echo.nasa.gov/api/ in January 2011. TRMM and MODIS-NDVI monthly
images from January 2002 to December 2010 were used. Those satellite images were
processed using ERDAS Imagine 9.1. The analysis was based on NDVI values by
investigating its increase after the construction of the trenches. In case of success, vegetation
growth should not only increase NDVI values, but also remain high throughout the year. An
independent two samples t-test was used to evaluate the impact of contour trenching to
vegetation growth. NDVI of areas with trench was compared with NDVI without trench. For
soil redistribution analysis, Cesium-137 ($^{137}$Cs) (Ritchie and McHenry, 1990, Zapata, 2003)
was used. By measuring the concentration of $^{137}$Cs in the vertical distribution, sources of
sediment can be identified (Walling and Quine, 1991; Wallbrink et al., 1999). We sampled
soil with a depth of 40-cm from the soil surface using split tube sampler, Eijkelkamp
Agrisearch Equipment, Giesbeek, The  Netherlands. Each point was sampled three times in a
radius of 1-m, mixed into one composite sample (Sutherland, 1994). In total, cesium
concentration of 128 samples was measured at the ISOLAB, Georg-August-Universitaet
Goettingen, Germany.
We conclude that the signal of greenness found was most likely due to alternating dry and wet
seasons, but does show short-term effect. Furthermore, TRMM is correlated to NDVI where
results show low correlation between TRMM and NDVI. The results of the erosion and
sedimentation analysis show the study area was previously already an eroded area. Sediments
found in the trench area are a combination of local and external sources. Early deposition
originates from local sources, followed by about 30-cm of sediments from external sources.
## 3.3  Indonesia Case: The potential of micro-hydro power plants on Maluku
islands, Indonesia
Indonesia has an abundance of water resources that can be used to create hydropower as a
valuable source of energy. This study is particularly focusing on Aboru village on Haruku
Island, where the project intended to build a micro-hydro power plant that could improve the
socio-economic situation of the local community (Balakrishnan 2006; Anyi et al. 2010). The
main objective of the research was to map locations with high-energy heads and assess the
minimum available annual discharges for potential micro-hydro in Aboru. The second
research effort aimed at finding potential locations for micro-hydro power plants on the
Maluku islands.





The Maluku islands are located in the eastern part of the Indonesia archipelago. In total, there
are 1027 islands. Most Maluku islands are mountainous (about 57%) and are mainly covered
by rainforests. The climate is humid, affected by monsoons and rainfall ranges from 1,000
mm to 5,000 mm year$^{-1}$. The study area is located in a small village with latitude 3° 35' 33" S
and longitude 128° 31' 0.7" E.
The potential of micro-hydro depends on two parameters, the river discharge and energy head.
The river discharge was measured uphill of the planned micro-hydro plant. Two divers
(Schlumberger Water Services, Delft, The Netherlands, measurements at 30-minute intervals)
were installed in the river to measure the pressure of surface water levels. To compare the
discharge results, a test using the dilution gauging method (Calkins and Dunne, 1970) was
also performed downstream several times at different locations. Furthermore, similar to a
study by Mosier et al (2012), a Digital Elevation Model (DEM) was used (see Fig.3). Data
was downloaded from http://srtm.csi.cgiar.org, which was provided by the Consortium for
Spatial Information of the Consultative Group for International Agricultural Research
(CGIAR).
We concluded that Maluku islands have a small potential for micro-hydro power plants.
Extrapolated discharges that can be used range from 0.03 m$^3$ s$^{-1}$ to almost 0.2 m$^3$ s$^{-1}$. As a
result, available streams with head conditions between 20 to 35 m can produce a minimum of
6 kW and maximum of 40 kW.
**4   Human actions towards intervention and hydrological research**
In an attempt to look at human actions towards intervention and hydrological research more
systematically, we started by identifying the process of participative actions from local
people. Typically, stakeholder involvement is perceived as – rather than communicating
things to people – seeking partnership in the process of (hydrological) change to affect
knowledge, attitudes and behaviour of participants in a project's network (Ertsen, 2002; see
also Poolman (2011) for a more extensive discussion about stakeholder participation in small-
scale water projects). Motivations for participation, including acts that may not necessarily be
seen as positive by other stakeholders, are a key component, especially when these change
over time – as we demonstrate below for our Vietnam case. However, there is little
recognition of motivations of individuals over time in the literature (see Cleaver (1999) and
Leahy (2008) for some attention to this issue).





For our three cases (see for the timeframes Table 1), we have drafted our own categorization
of human actions in intervention and hydrological research. In each of the interventions in
Vietnam, Kenya and Indonesia, local stakeholders of different kinds were engaged (see
Appendix A, Table A.1-A.6). This engagement included the hydrological research, especially
where it was part of the intervention itself. Our analysis will focus on community
participation, including what went differently than expected and the issue whether in the
future such developments could be anticipated upon. Based on the results, we develop
suggestions how hydrological researchers can include considerations on human agency when
planning and performing field research.
Human agency in intervention and research can be related to existing theories on community
participation. There are many participation theories; Arnstein (1969) introduced the ladder of
participation for urban development where the scale was from non-participation to being able
to make decision (citizen empowerment). The scale influenced other fields and was further
developed, for example by Choguill (1996). Her ladder of participation was based on the scale
of willingness of government in community projects. One recent participatory spectrum is
IAP2 (2007), where along the spectrum the impact of public participation increases. Another
participation framework during intervention phase was proposed by Srinivasan (1990), where
this was meant for training trainers in participatory technique. We found this last approach
useful in analysing our case studies, as the community participation scale from Srinivasan
(1990) allows for differentiating attitudes towards change, by sorting them along a scale
showing varying degrees of resistance or openness (see Fig. 4). Therefore, we found this
potential to "measure" attitude even more interesting because our results suggest that these
attitudes of stakeholders change over time, during the intervention and research itself.
As an example, we use the Vietnam case to gain an overview how the local community
participated in the intervention phase and how this altered over time. At the beginning of the
project, none of the landowners agreed with the intervention, especially because they had not
yet seen a successful example in their particular area. After negotiations, a monk organization
was willing to provide their land as an example case [#6A]. And after the large trenches were
constructed, the monk organization did not like the design. The rejection of the large trenches
enforced the proposer to reconsider the trench dimensions. Thus, the proposer came up with a
smaller design of contour trenches. Despite the smaller design, the monk organization still
refused to continue implementing the new design on its remaining land. Consequently, the





proposer introduced the smaller design to other farmers and one farmer accepted it. The
smaller trenches were then implemented in one farmers' area. The acceptance of the smaller
design by other farmers continued. Farmers living nearby requested also the small trenches to
be constructed on their land. After the monks' organization saw the results at several farmers'
land, the monk organization eventually requested the proposer to construct small trenches on
their remaining land. The decision of local people who wanted to have contour trenches
occurred after seeing an example of a smaller design. The implemented scale of the above
community participation (Srinivasan, 1990) can be seen in Fig. 5 where:
• 0 to 6; among many landowners, only a monk organization was willing "to try some
actions" on their land.
• 6 to 3; the monk organization was sceptic and "have doubts" if they continue
implementing the trenches, even with a smaller design.
• 3 to 6; a farmer was willing "to try some actions" on his land.
• 6 to 7; from one farmer with smaller trench design, he advocated change so that the
acceptance of the smaller design continued in this particular area.
Since we were not involved during the intervention phase in the Kenya and that the Indonesia
case resulted to cancelation of field intervention, only the community participation in
hydrological research will be analysed in the remainder.
**Categorization of human actions in hydrological field research**
Within the context where intervention was done simultaneously with hydrological research –
the Vietnam case [#6] – the actual shape of the final intervention was decided upon within
several rounds of discussions between project team and local communities. Agreement was
obtained through a negotiation process. The actual shape of the hydrological research was
heavily dependent upon knowing the definitive location of the intervention. While the
decision process took place, measurements were conducted in the vicinity of the possible
locations of the intervention. On-site measurements had to be re-evaluated from time to time
due to changes of intervention locations. The intervention period and financial support for
research were both limited and limiting as well. Most likely, in conditions of simultaneous
intervention and research, changes required adjustments to a new setup, which often means





increasing financial expenditure for measurements. Therefore, any decision to start either
intervention or hydrological research needs careful thought.
In general, we find different processes of involvement and different human actions related to
the three hydrological research projects (see Table 4a, on events labelled with [#]). For
example, in Vietnam and Kenya, access tubes [#3, #9A] were taken away. Also, in Vietnam
the Divers were taken away [#4]. In Kenya, one rain gauge was damaged by elephants, and
afterwards removed by local people [#7]. Next to human agency affecting the hydrological
research, other events affected the research activities. In Vietnam, one rain gauge clogged
[#1] because of fine sands from strong winds, and the screen of the observation wells [#5]
proved to be not suitable for local conditions. Obviously, these events could have been
avoided. Rain gauges could have been checked and maintained on a regular basis, especially
when realizing that local conditions and climate might affect the measurement. When
planning to conduct isotope analysis, observation well structures should have been
constructed for a proper sampling. However, there were also problems that probably could not
have been avoided, especially technical failures of data loggers [#2, #8, #10, #11] from
tipping buckets and divers.
Table 2b also provides the detailed results in terms of timing and type of human actions
during intervention processes. For example, the Vietnamese intervention could only be
constructed after many negotiations between the proposer and the end users. Such a decision
could change the final location of the intervention, which in turn affected directly the
hydrological research. In the Vietnam case, intervention design and location were determined
by the local people, who had the power to choose their preference of intervention and decided
whether it could be implemented on their land or not. In the Kenyan case, intervention design
and location were simply accepted by the local Maasai and decisions were made by KWS. In
this case, the intervention existed first and was evaluated later. In addition, negotiating about
reasonable labour costs for the field study in 2010 resulted in lack of local assistance for soil
moisture measurements. In the Indonesian case, the intervention was not, as was preferred
before, an outcome as a recommendation from the hydrological research. The end user of the
intervention shifted from a pilot at a village to a micro hydro model at a local university. The
intervention was cancelled due to insufficient funding, even when the hydrological research
went smoothly.





The Srinivasan scale allows for analysing the changes in attitudes and possible actions
concerning an intervention over time. However, the scale seems to be less relevant for the
hydrological research itself, which is actually interesting as it suggests that stakeholders may
have different attitudes and ideas with respect to interventions. To what extent this motivation
is always directly linked to an attitude towards the intervention, remains an open question. An
example is the Vietnam case, where access tubes and divers were taken away. Possible
reasons are that someone rejected the project, did not want any intervention to be constructed
on the land, had negative impressions of the intervention, or was not satisfied with the
proposer's offer. On the other hand, the attractiveness of the device itself and/or curiosity
could make people eager to have such devices. Therefore, the resulting human action to
remove the device may not have been a rejection of the project at all, but just a desire to own
a device with a unique appearance.
**5   Planning for surprises in hydrological research**
In all our three case studies, we conducted different measurement techniques depending on
the research objectives. What all case studies had in common was that the projects had to be
changed due to local negotiations. No matter the scale of either stakeholders' participation in
hydrological research or their motivations, one will have to face human actions –
disappearance of measurement devices, changes of locations, etcetera – when designing field
research. The events could have been anticipated – or even (partially) avoided - but usually
were treated as surprises or unforeseen side-effects. Learning from our own experience, we
claim that they should at least be anticipated. For example in the Vietnam case, when the
divers disappeared, a stronger cover for the observation wells might have been used. In the
Kenya case, a more secure location for some devices could have been prepared to cope with
communities outside the research area ("third party surprises"). The RAND studies provide
guidance for an approach that anticipates on known surprises (Dewar, 2002). In planning for
surprises, as outcomes of local negotiations are not known before, we envision that a
hydrological field researcher prepares the study by taking into account several scenarios.
Thinking in scenarios for hydrological fieldwork instead of one single approach allows for
making decisions based on expected implications of events on the hydrological results, and
should minimize the costs of improvisation.
We developed three research budget scenarios for the three cases, where we defined
effectiveness in terms of process understanding and important model input. First, we



evaluated the technical approaches per case study (see Table 2-4) in terms of performance
(Blume et al., 2008), which is the effectiveness of measurements in understanding
hydrological processes. Then, expenditures included in our (fictitious) budgets are labour and
financial costs, which are shown in ranges of Euros; (+-) is between 0 to 50 Euro, (+) 50 to
250 Euro, (++) 250 to 750 Euro, and (+++) above 750 Euro. These ranges are given as
examples to illustrate the expenditures.
Either collecting more data and/or different data is usually the choice we have to make to
confirm certain underlying dominant hydrological processes due to an intervention. We used
cost-benefit analysis (Sassone, 1978) in research scenarios that were developed based on the
Delphi method (Linstone and Turoff, 1975). Each scenario specifies a budget; the
measurements that can be conducted within that budget, and the dominant hydrological
processes studied. In changing the budgets, we could explore changes in and differences
between probable field campaigns, especially in gaining better understanding of dominant
mechanisms of the intervention.
## 5.1   Scenarios
We tested the scenario approach with a group of experts. We offered three scenarios. Scenario
1 was approximately at the lowest budget, which was estimated by considering the
experiences gained by the author. As it was already known how the research went, the lowest
cost scenario was drafted by eliminating the measurements that failed or were not used in the
analysis. This combined at least a desk study with field measurement data. Also, this was a
theoretical baseline scenario for good understanding of the intervention.
Scenarios 2 and 3 covered a longer research period. Extension of measurement and
performing other methods were proposed. There were several options related to parameters
that were selected and added, with various spatial and temporal combinations. Those options
were:
A. Extension of the measurement period.
B. Additional samplings.
C. Additional measurement devices.
D. Additional analysis.



Option C and D are connected since having another type of measurement might use the same
or require a new (commercial) software program or service.
Scenario 2 was included a budget increase of about 20%. Options for an extension of the
measurement period and more samplings were preferred.
Scenario 3 used an approximately 80% budget increase. It implies a condition with an
expansion of the second scenario combined with much more room for additional parameters
in the field campaign.
Some assumptions for the budgeting were set as follows:
• Related research budget components like field personnel, transportation to the site,
meals, and accommodation were not considered.
• A researcher was categorized as non-paid labour in the research area, since s/he
receives salary from the researcher's institution. Thus, the researcher's expenses were
ignored.
• Shipping cost of devices and samples, taxes of research devices, and research permit
costs were excluded.
• There were no subsidies from research institutions for measurements devices or
models.
• None of the scenarios took into account decisions made for a particular intervention
and its development.
For Scenarios 2 and 3, the end results of possible field campaigns and analysis were discussed
with ten experts from different Dutch institutions, who were selected from the working
environment of the author. Each scenario had its own specific hydrological objective that fits
to an expertise (i.e. hydro-geology, hydrology, remote sensing), but the experts had different
hydrological backgrounds. The implemented research with the results and proposed scenarios
of several field campaigns were explained to the experts to clarify the content and objective of
the research. Subsequently, they had to grade the scenarios based on the level of additional
understanding (if any) that would be achieved. The required budget itself was not mentioned
to allow experts to objectively value the proposal without any economic consideration. The
author picked the Dutch grading scale as follows:
• 1-5,5 = little understanding of the relevant mechanism of intervention





- 6-7,5 = good understanding of the dominant mechanism of intervention

- 8-8,5 = better understanding of the dominant mechanism of intervention

- 9-10 = complete (full process) understanding of the mechanisms relevant for the intervention

In the last part of the interview, the experts were also given the opportunity to provide their own alternative approaches that could result in better understanding.

Even though this was a theoretical exercise and that it was not easy to provide clear-cut evidence for the scenarios to be realistic enough, results are useful. There may be many other options of optimization, such as cheaper measurement devices and modelling. Different research institutions prefer different measurement devices, or software developed by certain institutions. Research institutions might already own measurement devices and software, thus do not want to spend money on others. This specific setup is merely an estimation in the context of the three case studies and may well vary from person to person due to people's preference. However, by asking ten experts for their input and analyze further their responses over the entire width of the scenarios, a good degree of objectivity, certainty and reality can be reached, if not in absolute, then at least in comparative terms. Our results are given in Fig. 6. We discuss the Vietnam case in more detail.

## 5.2 Vietnam case

The lowest budget for having sufficient understanding of groundwater recharge gained in the actual research is reduced to almost 70% of the expenses during implementation (see Table B.1). Rainfall measurement is a must for the input of the model. The hydraulic properties of soil and infiltration tests are important as well. The water level measurement is required to get the ponding in the trench correctly. These costs are not much compared to other measurements. Soil moisture measurement is removed from the field campaign since it is not only expensive, but also the access tubes are prone to be taken away by the local people. Isotope tracers are excluded, because the constructed observation wells were not suitable for groundwater sampling. In addition, the cost for this analysis is considered to be expensive. On the other hand, isotopes are beneficial and will provide signals as long as the observation wells would be better constructed. A minimum of three observation wells are set, since it is the minimum or triangle layout to get an idea on the groundwater flow direction. A short but



sufficient period of measurements would be during the wet season, where the trench may be
filled with rain water.
Even though the cost reduction is significant, the conditions to apply these methods could
remain uncertain. For example, when a researcher made a plan for scheduling the starting
point of measurement at the beginning of a wet season, no one would expect at first that
negotiating with the local community was difficult, even though this determines whether or
not the intervention can be built or continued. There has to be willingness from the
community to provide land for the intervention. After several discussions and meetings, a
local to local approach was needed to convince stakeholders that the intervention would be
beneficial to the local community. However, no one could predict when and where it could be
realized. If the decision to be made for construction was delayed, the plan for hydrological
measurements would have to wait until the next wet season, which would have been one year
later. And if there is a tension to install the measurement devices for a "with and without"
analysis, and the location shifts in time, new measurement set ups have to be adjusted. These
conditions will result in loss of data and time for the hydrological research. As such, the
minimum budget is somewhat artificial. The other way around, the big difference between the
minimum budget and the actual budget suggests that in the Vietnam case, negotiations on the
intervention brought along high costs.
When more budget would be available, scenario 2 (see Table B.2) could expand the
implemented program by constructing one new observation well and its groundwater level
measurements. Also the sampling period for isotope tracer is added. The observation well
should be placed in line with the existing wells and its screen should be along the pipe, from
near soil surface to the bedrock. It would be expected that the recharge can be more apparent
where the signal of infiltrated rainwater can directly infiltrate into the pipe. Thus, the
groundwater fluctuation and sampling can confirm the result of the implemented research.
In scenario 3 (see Table B.3), an 80% increased budget gives options for more applications
and/or more advanced methods. Besides one new observation well and isotope samplings,
three other wells should be constructed. The observation wells should be placed at the small
trench area. A possible advanced measurement is by performing an Electrical Resistance
Tomography (ERT) survey for subsurface imaging. Several cross sections of the subsurface
could be obtained during the dry and wet period. By having these new wells combined with





the analyzed ERT data, the hypotheses could be made more pronounced regarding the
difference in groundwater behaviour with and without the intervention structure.

### 5.3  Interviews with experts

The results of the interviews with the experts can be seen from Table C.1-C.3. Of the three
cases, the Vietnam case had most options, due to better financial conditions, compared than
the other two cases. Considering scenario 2, 70% of the experts believe an additional well and
a 1-year continuation of the groundwater level measurements, including isotope samplings
and analysis, would result in similar data collection as in the implemented research. One
expert considered that extra data might even lead to confusion. Another period of 1-year data
could be used for validation, thus might give more confidence. A very long data series, from
two to about ten years of groundwater level measurement would be very beneficial for better
understanding the mechanism of the recharge. In the Kenya case, 60% of the experts value the
outcomes of additional soil moisture measurement, extension of rainfall and NDVI images as
similar to the implemented research. The remaining 40% think that new soil moisture
measurements could lead to additional understanding. For Indonesia, 90% of the experts think
that the result of extending discharge measurement will not increase understanding. However,
one expert says new data matter, as measurements could have been made during a very dry or
very wet year.
In Scenario 3, with 80% increase in budget, the value of measurements points to similar
results as in Scenario 2, with some additional elements. For Vietnam, 80% of the experts say
that Electrical Resistivity Tomography (ERT) measurements could increase the understanding
of mechanism of the recharge and provide more explanation of the disconnected groundwater
system. Thus, it could potentially confirm the groundwater profile and the groundwater level
during recharge. Performing ERT either during dry or wet seasons sometimes yields results
hard to interpret, since ERT is a static measurement. In the Kenya case, 70% of the experts
say adding higher resolution of 10 m might be sufficient to capture the greenness of the
trenches. The images could be of importance to see a hypothetically constant greenness
signal. Finally, for the Indonesia case, as a micro-hydro installation usually requires a
minimum annual discharge, a long-term discharge will be used for discharge statistics.
Finally, 60% of the experts believe that more discharge measurements at different locations
with different soil types, geology, and land uses, could improve our understanding, especially
the discharge response of different catchments on different islands.



In summary, a research plan with 20% increased funding (Scenario 2) appears to obtain similar understanding as the reference result. On the other hand, an 80% increase in funding may be capable of gaining a better understanding. A costly research plan for a small-scale intervention project may not be economically feasible and thus impossible to implement.

## 6    Towards systematic planning

Despite all the problems we encountered in the three field research projects, we could develop a good understanding of the hydrological impacts of interventions in three different developing countries. In Vietnam, during the wet season, contour trenches contribute to recharge, but only for short-term impact, up to two months. In Kenya, vegetation growth in the trench area as reflected in the signal of greenness index was most likely due to the wet season, without a clear long-term effect from the trenches. In Indonesia, the potential of micro-hydro capacity on Maluku islands ranges from 6 to 40 kW. In the three cases local people participated during the implementation of the projects, both in the intervention and hydrological research. As a result, the field campaigns were not perfect in terms of hydrological standards. Measurement devices were damaged, removed, disappeared or not located at the final intervention. In the end, we ended up with less data or data of lower quality. Local participation and financial constraints forced us to deal with research and intervention as interacting with and affecting each other.

As this setting is not unique to our three small cases, balancing intervention and research is a general challenge. Tracing back the social reality and the way it shapes research and intervention with the associated budget allowed us to gain more insight into trade-offs between hydrological knowledge and hydrological research management. Based on our experiences, we propose that planning ahead is possible. We propose a new systematic perspective on how to prepare hydrological research for a more effective way to implement small-scale water intervention research projects. Being prepared for and responsive to surprises due to human actions can be achieved by developing scenarios that combine hydrological issues with cost-benefit analysis. Considering financial costs and specific research objectives of small-scale interventions, options for field campaigns and analysis that could answer the research questions can then be defined.

Baiocchi and Fox (2013) suggest six key issues to be prepared for and respond to surprises; (1) learn from experience: attract and retain the most experienced people, (2) address the negative effects of surprise, (3) assess the level of chaos in the work environment, (4) prepare





for "third-party surprises", (5) focus on building a network of trusted colleagues, and (6)
conduct regular future-planning exercises. Their recommendations confirm our ideas:
planning for surprise requires proper understanding of small interventions within their
hydrological context and incorporating interdisciplinary knowledge, learning, and local
participation (see Karjalainen et al., 2013; Rodela et al., 2012; Reed et al. 2010).
Similar to balancing development and conservation (Garnett et al., 2007), when financial
constraints – and usually time as well – become important, a researcher should be able to
balance what he/she can and cannot do. Since budgets and time for a small-scale intervention
are limited, research should be well planned. In order to include the costs of performing
hydrological studies and the efficiency (effectiveness) in planning for surprises, we discussed
an approach applying cost-benefit analysis. Despite its simplicity, it appears to be a good way
to quantify research efforts versus the (probable) outcomes.
The judgments of the outcomes were obtained from interviews with water experts. Sharing
options with other experts adds value to the preparation. Each scholar has their own
preferences, and thus there is no single solution. This was shown during the interviews with
the experts, when they were forced to make a choice by pushing their preference in grading
the available field campaign options. Eventually, even when incorporating experts' inputs, we
will still have to make decisions and will possibly select our own preferred choices. In the
end, dealing with the local constraints is a decision to be made by the researcher. However, by
doing the two analyses of scenarios and cost benefits continuously during planning and
performing hydrological research, one will be better informed to make decisions.
The notion that the effects of human actions to be expected in hydrological field campaign are
basically unspecified does not imply that they could not adequately and fruitfully be
translated in specific planning, as we have shown. Taking into account human actions in
planning field campaigns for something that is usually seen as a single-scientific activity
implies that each field design should be tuned to the situation under consideration. A designer
cannot come up with a standard solution and may experience different stages of learning
processes that continue to shape both intervention and hydrological research (see Fig. 7).
Paradoxically, introducing such a multifaceted approach asks for hydrological researchers
with higher qualifications. Planned improvisation needs scientific expertise, as much as it
requires a specific attitude.





**Appendix A: Participative Actions from Local People**
**Appendix B: Cost Scenarios**
**Appendix C: The experts' opinions**
**Acknowledgements**
The authors would like to thank the funding agencies and key people for their support in each
of the three small-scale water projects; Royal Haskoning Vietnam, Marieke Nieuwaal, and
local partners, both from the community and the Vietnamese government; International
Foundation for Science, Sweden, Dr. Cox Sitters (Moi University, Kenya), Dr. Jeannis-Nicos
Leist (University of Goettingen); the Dutch Ministry of Economic Affairs, Agriculture, and
Innovation, and the project consortium (Noes Tuankotta, UKIM and IBEKA). We also would
like to thank the local people who helped us in the field for technical and logistic assistance,
but are not mentioned here one by one. Lastly, we thank the ten water experts for their
participation in the interviews.



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



Table 1. Time periods of three hydrological intervention-based research projects

| Case study | Intervention | | Hydrological research | |
|---|---|---|---|---|
| | Start | End | Start | End |
| Vietnam | October 2007 | September 2008 | October 2007 | March 2011 |
| Kenya | 2002 | 2003 | September 2010 | March 2012 |
| Indonesia | September 2012 | September 2013 | July 2010 | October 2011 |

Table 2. Evaluation of technical approaches: gain versus expenditure Vietnam case

| Parameter | Method | Gain | | Expenditure | | Problems |
|---|---|---|---|---|---|---|
| | | Process | Model | Labour | Cost | |
| Rainfall | Tipping bucket | ++ | + | +- | +++ | Clogging due to fine sand & logger |
| Soil moisture | TDR (with 8 access tubes) | +- | +- | ++ | +++ | Prone to vandalism |
| Vertical flow path | Dye tracer | ++ | ++ | +- | +- | Destructive sampling |
| Soil physics | Lab analysis | ++ | ++ | +- | + | Point data, lack of deeper samples |
| Infiltration capacity | Inverse auger test | ++ | ++ | +- | + | Point data |
| Water level | Meter height reading | +++ | +++ | +- | +- | Point data |
| Groundwater level | Observation well (reached bedrock depth) | +++ | +++ | + | +++ | Point data & divers prone to vandalism |
| Isotope tracer | Lab analysis | +++ | ++ | + | +++ | Short period of sampling |

Notes:
(+) positive rating = greater gain

Table 3. Evaluation of the technical approaches: gain versus expenditure Kenya case

| Parameter | Method | Gain | | Expenditure | | Problems |
|---|---|---|---|---|---|---|
| | | Process | Model | Labour | Cost | |
| Rainfall | Tipping bucket | ++ | + | + | +++ | Logger & removal |
| | TRMM (remote sensing) | ++ | ++ | - | - | Low resolution |
| Soil moisture | TDR (with 6 access tubes) | +- | +- | +++ | +++ | Point data & prone to vandalism |
| Soil physics | Lab analysis | ++ | ++ | + | +- | Sample composition |
| Greenness index | NDVI (remote sensing) | ++ | ++ | - | - | Low resolution |
| Erosion & sedimentation | Cs analysis | ++ | ++ | + | +++ | Reference point |

Notes:
(+) positive rating = greater gain

Table 4. Evaluation of the technical approaches: gain versus expenditure Indonesia case

| Parameter | Method | Gain | | Expenditure | | Problems |
|---|---|---|---|---|---|---|
| | | Process | Model | Labour | Cost | |
| Rainfall | Tipping bucket | ++ | ++ | +- | +++ | Logger |
| Discharge | Velocity area | ++ | ++ | +- | +++ | Logger |
| | Dilution gauging | ++ | ++ | +- | +- | Short measurement |
| Head | DEM (remote sensing) | ++ | ++ | - | - | Low resolution |

Notes:
(+) positive rating = greater gain





1    Table A.1. Vietnam case; hydrological research

| **Procedure** (the official way of conducting hydrological research) | **Event** or **observation** (an indisputable happening) | **Action** (the process before an event) | **Interpretation** (giving a meaning of the event and/or action to research) | **Period** |
|---|---|---|---|---|
| Install two rain gauges | Intermittent rainfall data | - | - | October 2007-March 2011 |
| | Clogged rain gauges [#1] | Need of checks and maintenance | Loss of rainfall data series | October 2008 |
| | Logger failed to record events [#2] | Tried to retrieved data from manufacture company and manual measurement | Loss of rainfall data series | December 2009-March 2011 |
| Install access tubes for soil moisture measurements | Loss of access tubes [#3] | Substituted with new access tubes | Loss of soil moisture data series, extra costs for new tubes | September 2008-March 2011 |
| Check infiltration at the bottom of the trench | - | - | - | September 2009- November 2009 |
| Measure soil porosity and bulk density | - | - | - | October 2007-June 2009 |
| Measure infiltration capacity | - | - | - | October 2007 & April 2009 |
| Measure surface water level in the trenches during wet season | - | - | - | October 2007 - November 2009 |
| Construct observation wells | Loss of divers [#4] | Perform manual measurement | Loss in groundwater level data series | December 2007 |
| | Improper screen instalment for isotope sampling [#5] | Nothing | Possible misinterpretation of isotope signal | October 2007 & April 2009 |
| | Construction before intervention [#6] | Extra costs for constructing new wells | A shift in location of intervention requires more measurements, thus more cost | October 2007-April 2009 |





1    Table A.2. Vietnam case; intervention

| **Procedure** (the common way of intervention) | **Event** or observation (an indisputable happening) | **Action** (the process before an event) | **Interpretation** (giving a meaning of the event and/or action to intervention) | **Period** |
|---|---|---|---|---|
| Introducing the concept of intervention to local authority and community | Meetings | Presentations and discussions to obtain support | Needed an agreement from local community | May 2006 - September 2007 |
| Constructing contour trenches [#6A] | A monks' organization provided their land in a size of 12 ha for intervention | Larger trenches were constructed on 8 ha area | Needed an agreement on someone's land to be interfered | October 2007 |
| | After large contour trenching, the monks' organization refused to continue construction on their remaining land | Meetings and discussions to convince the intervention would be beneficial for the community | Although the monks gave permission for large contour trenching, they did not like the design | November 2007 - March 2008 |
| | The proposer approached other land owners | The proposer offered a smaller contour trench design | Negotiated on the design | March 2008 |
| | A local farmer excepted the smaller trench design | Construction of smaller contour trenches on 1 ha area | The smaller design was tested | April 2008 - May 2008 |
| | Other local farmers requested smaller contour trenching. | Construction of smaller contour trenches on other farmers' land (10 ha) | The smaller design was preferred | June 2008 - August 2008 |
| | The monks' organization also provided their remaining land for smaller contour trenching | Construction of smaller contour trenches on the remaining 4 ha area. | Overall in this particular local community, a larger design was not accepted. | June 2008 - August 2008 |



1    Table A.3. Kenya case; hydrological research

| Procedure (the official way of conducting hydrological research) | Event or observation (an indisputable happening) | Action (the process before an event) | Interpretation (giving a meaning or impact of the event to research) | Period |
|---|---|---|---|---|
| Install two rain gauges | One rain gauge was damaged by elephants and thus removed by local people [#7] | Information came very late, thus arrangement for reset up of the rain gauge could not be performed | Loss of rainfall data series | September 2010-March 2012 |
| | One logger failed to record events [#8] | Tried to retrieve data without success | Loss of rainfall data series | September 2010-March 2012 |
| Soil moisture measurements to be conducted by a local person | A long negotiation to start measurement was not successful [#9] | Established new connection with other local people was not successful too | Loss of soil moisture data series | September 2010-Mach 2013 |
| | Loss of access tubes [#9A] | Installed two remaining tubes | Loss of soil moisture data series | September 2010-March 2012 |
| Used TRMM & NDVI analysis | - | - | - | January 2011-March 2011 |
| Measure soil porosity and bulk density | - | - | - | September 2010 |
| Soil sampling for Cesium analysis | - | - | - | September 2010 |

3    Table A.4. Kenya case; intervention

| Procedure (the common way of intervention) | Event or observation (an indisputable happening) | Action (the process before an event) | Interpretation (giving a meaning of the event and/or action to intervention) | Period |
|---|---|---|---|---|
| Introducing the concept of intervention to local authority and community | Meetings | Convincing local people with success | Local people accepted the design | 2001-2002 |
| Constructing contour trenches | The majority of Maasai supported contour trenches | Trenches were first constructed in smaller dimension and furthermore in larger ones | Easy to implement different dimension of contour trenching in this particular area | 2002-2006 |
| After construction of large contour trenches | - | - | - | 2002-present |





1    Table A.5. Indonesia case; hydrological research

| **Procedure** (the official way of conducting hydrological research) | **Event** or observation (an indisputable happening) | **Action** (the process before an event) | **Interpretation** (giving a meaning or impact of the event to research) | **Period** |
|---|---|---|---|---|
| Install two rain gauges | One logger failed to record events [#10] | Only counted on one logger | Loss of rainfall data series | July 2010-July2011 |
| Install two divers | One logger failed to record events [#11] | Only count on one diver | Loss of water level data series | July 2010-July 2011 |
| Measure discharge using dilution method and velocity area | - | - | - | February 2011-March 2011 |
| Used DEM analysis | - | - | - | April 2011-June 2011 |

3    Table A.6. Indonesia case; intervention

| **Procedure** (the common way of intervention) | **Event** or **observation** (an indisputable happening) | **Action** (the process before an event) | **Interpretation** (giving a meaning of the event and/or action to intervention) | **Period** |
|---|---|---|---|---|
| Proposed intervention to local authority and community | Meetings, permit issue, estimation of micro hydro budget and research on its potential | - | - | March 2010-June 2011 |
| Design suitable micro hydro installation | Two plans were agreed; first an installation of about 80kW and second small kW was estimated after the research | Search for extra funding to meet the construction cost | A decision had to be made based on the availability of funding | July 2010-January 2012 |
| Pilot result | Research result suggests little potential for micro-hydro installation in a village, but funding was still not enough | Constructed micro-hydro model for a local university | The final intervention shifted from a pilot to a model | September 2012-September 2013 |





Table B.1. Vietnam Case, Scenario 1: to measure rainfall and groundwater level for a short
period

| Parameter | Method | Number / Samples | Labour | In Euro | Cost | In Euro |
|---|---|---|---|---|---|---|
| Rainfall | Tipping bucket | 2 | +- | 40 | +++ | 2.015 |
| Soil physics | Lab analysis | 6 | +- | | + | 238 |
| Infiltration capacity | Inversed auger test | 8 | +- | | + | 75 |
| Water level | Meter height reading | 13 | + | | +- | 15 |
| Groundwater level | Observation well & diver (reached bedrock) | 3 | ++ | 720 | +++ | 5.533 |
| | | | **Total I** | **760** | **Total II** | **7.876** |

Notes:
(+-) = 0 - 50 Euro, (+) = 50 - 250 Euro, (++) = 250 - 750 Euro, (+++) above 750 Euro
Table B.2. Vietnam Case, Scenario 2: to recheck the signal of recharge

| Parameter | Method | Number / Samples | Labour | In Euro | Cost | In Euro |
|---|---|---|---|---|---|---|
| Rainfall | Tipping bucket | 2 | +- | 40 | +++ | 2.015 |
| Soil moisture | TDR (with access tubes) | 8 | +++ | 2.160 | +++ | 4.717 |
| Vertical flow path | Dye tracer | 3 | +- | | +- | 10 |
| Soil physics | Lab analysis | 6 | +- | | + | 238 |
| Infiltration capacity | Inversed auger test | 8 | +- | | + | 75 |
| Water level | Meter height reading | 13 | +- | | +- | 15 |
| Groundwater level | Observation well & diver (reached bedrock) | 7 + 1 | ++ | | +++ | 13.893 |
| Isotope tracer | Lab analysis | 116 + 116 | + | | +++ | 1.856 |
| | | | **Total I** | **2.200** | **Total II** | **22.819** |

Notes:
(+-) = 0 - 50 Euro, (+) = 50 - 250 Euro, (++) = 250 - 750 Euro, (+++) above 750 Euro





Table B.3. Vietnam Case, Scenario 3: to map the subsurface

| Parameter | Method | Number / Samples | Labour | In Euro | Cost | In Euro |
|---|---|---|---|---|---|---|
| Rainfall | Tipping bucket | 2 | +- | 40 | +++ | 2.015 |
| Soil moisture | TDR (with access tubes) | 8 | +++ | 2.160 | +++ | 4.717 |
| Vertical flow path | Dye tracer | 3 | +- | | +- | 10 |
| Soil physics | Lab analysis | 6 | +- | | + | 238 |
| Infiltration capacity | Inversed auger test | 8 | +- | | + | 75 |
| Water level | Meter height reading | 13 | +- | | +- | 15 |
| Groundwater level | Observation well & diver (reached bedrock) | 7 + 4 | + | | +++ | 18.909 |
| Isotope tracer | Lab analysis | 116 + 116 | + | | +++ | 928 |
| Subsurface mapping | ERT | 4 | | 2.000 | +++ | 10.000 |
| | | | **Total I** | **4.200** | **Total II** | **36.907** |

Notes:
(+-) = 0 - 50 Euro, (+) = 50 - 250 Euro, (++) = 250 - 750 Euro, (+++) above 750 Euro
Table B.4. Kenya Case, Scenario 1: to use remote sensing data

| Parameter | Method | Number / Samples | Labour | In Euro | Cost | In Euro |
|---|---|---|---|---|---|---|
| Rainfall | Tipping bucket | 2 | + | 120 | +++ | 1.305 |
| | Remote sensing analyis | | - | | - | |
| Greenness index | Remote sensing analysis | | - | | - | |
| | | | **Total I** | **120** | **Total II** | **1.305** |

Notes:
(+-) = 0 - 50 Euro, (+) = 50 - 250 Euro, (++) = 250 - 750 Euro, (+++) above 750 Euro
Table B.5. Kenya Case, Scenario 2: to retry one year soil moisture measurement

| Parameter | Method | Number / Samples | Labour | In Euro | Cost | In Euro |
|---|---|---|---|---|---|---|
| Rainfall | Tipping bucket | 2 | + | 120 | +++ | 1.305 |
| | Remote sensing analysis | | - | | - | |
| Soil moisture | TDR (with access tubes) | 6 | +++ | 2.880 | +++ | 4.990 |
| Soil physics | Lab analysis | 8 | +- | 40 | +- | |
| Greenness index | Remote sensing analysis | | - | | - | |
| Erosion & sedimentation | Cs analysis | 128 | + | 100 | +++ | 1.699 |
| | | | **Total I** | **3.140** | **Total II** | **7.994** |

Notes:
(+-) = 0 - 50 Euro, (+) = 50 - 250 Euro, (++) = 250 - 750 Euro, (+++) above 750 Euro



Table B.6. Kenya Case, Scenario 3: to maximize remote sensing analysis

| Parameter | Method | Number / Samples | Labour | In Euro | Cost | In Euro |
|---|---|---|---|---|---|---|
| Rainfall | Tipping bucket | 2 | + | 120 | +++ | 1.305 |
|  | Remote sensing analysis |  | - |  | - |  |
| Soil moisture | TDR (with access tubes) | 6 | +++ | 2.880 | +++ | 4.990 |
| Soil physics | Lab analysis | 8 | + | 40 | +- |  |
| Greenness index | Remote sensing analysis |  | - |  | - |  |
| Erosion & sedimentation | Cs analysis | 128 | + | 100 | +++ | 1.699 |
| High resolution greenness index | Remote sensing analysis |  | - |  | +++ | 8.400 |
|  |  |  | **Total I** | **3.140** | **Total II** | **16.394** |

Notes:
(+-) = 0 - 50 Euro, (+) = 50 - 250 Euro, (++) = 250 - 750 Euro, (+++) above 750 Euro
Table B.7. Indonesia case, Scenario 1: to measure discharge of one river for one year

| Parameter | Method | Number / Samples | Labour | In Euro | Cost | In Euro |
|---|---|---|---|---|---|---|
| Discharge | Velocity area (diver) | 3 | +- |  | +++ | 1.500 |
|  | Dilution gauging |  | +- |  | +- | 25 |
| Head | Remote sensing analysis |  | - |  | - |  |
|  |  |  | **Total I** | **50** | **Total II** | **1.525** |

Notes:
(+-) = 0 - 50 Euro, (+) = 50 - 250 Euro, (++) = 250 - 750 Euro, (+++) above 750 Euro
Table B.8. Indonesia case, Scenario 2: to investigate discharge of another river

| Parameter | Method | Number / Samples | Labour | In Euro | Cost | In Euro |
|---|---|---|---|---|---|---|
| Rainfall | Tipping bucket | 2 | +- | 50 | +++ | 1.990 |
| Discharge | Velocity area (diver) | 3 + 2 | +- |  | +++ | 2.500 |
|  | Dilution gauging |  | +- |  | +- | 25 |
| Head | Remote sensing analysis |  | - |  | - |  |
|  |  |  | **Total I** | **50** | **Total II** | **4.515** |

Notes:
(+-) = 0 - 50 Euro, (+) = 50 - 250 Euro, (++) = 250 - 750 Euro, (+++) above 750 Euro





1    Table B.9. Indonesia case, Scenario 3: to investigate discharges of four other rivers

| Parameter | Method | Number / Samples | Labour | In Euro | Cost | In Euro |
|-----------|--------|------------------|--------|---------|------|---------|
| Rainfall | Tipping bucket | 2 | +- | 50 | +++ | 1.990 |
| Discharge | Velocity area (diver) | 3 + 8 | +- | | +++ | 5.500 |
| | Dilution gauging | | +- | | +- | 25 |
| Head | Remote sensing analysis | | - | | - | |
| | | | **Total I** | **50** | **Total II** | **7.515** |

3    Notes:
4    (+-) = 0 - 50 Euro, (+) = 50 - 250 Euro, (++) = 250 - 750 Euro, (+++) above 750 Euro



1   Table C.1. Vietnam case; the experts' opinions

| No | Title | Institution | Scenario 2 | | Scenario 3 | | Other suggestions |
|---|---|---|---|---|---|---|---|
| | | | Grade | Remarks | Grade | Remarks | Remarks |
| 1 | PhD | Utrecht University | 7 | Seasonality is already included | 7 | Disadvantage: profiles measured only one time | Require 10 year groundwater level data |
| 2 | MSc | Delft University | 8 | Measure for at least 2 years | 8.5 | Model only tests hypothesis. Measurements already answered the research question | - |
| 3 | MSc | Delft University | 7 | Isotope is an advance method with good result | 7 | One time measurement equals to nothing | To study the unsaturated zone, to measure rate of recharge using SM sensors etc |
| 4 | PhD | Delft University | 8 | Need validation and try to get more confidence or to decrease uncertainty. But it could even lead to confusing results | 8 | Hard to interpret | Depending on Ks and soil moisture. Challenging (qualitative result): infiltration test and surface water measurements |
| 5 | PhD | Delft University | 7 | Sceptic | 8 | - | - |
| 6 | Prof | UNESCO-IHE | 7.5 | - | 8 | - | - |
| 7 | PhD | UNESCO-IHE | 7.5 | - | 8 | Increase resistivity of water by injecting sodium chloride | More artificial tracer, (yellow dye), soil moisture measurement below the trench (use cheap sensors like Decagon). A need of timely scale measurements or time laps measurements |
| 8 | PhD | Delft University | 8 | - | 8 | - | Previous measurements were already sufficient |
| 9 | MSc | Eindhoven-Deltares | 7 | - | 8 | - | It would be an advantage to have 3-D |
| 10 | PhD | Delft University | 7.5 | - | 8 | Expensive (cost magnitude about 10.000 to 30.000 Euro for a 5 m interval) | Geophysical approach for spatial information. Soil type analysis, ground radar method, and 1-2 points tracer (pollution) |



1    Table C.2. Kenyan case; the experts' opinions

| No | Title | Institution | Scenario 2 | | Scenario 3 | | Other suggestions |
|----|-------|-------------|------------|---------|------------|---------|-------------------|
| | | | Grade | Remarks | Grade | Remarks | Remarks |
| 1 | PhD | Utrecht University | 7 | NDVI related to temperature, LAI, but does not correlate to soil moisture | 7 | - | Try FPAR (Fractional Photosynthetically Active Radiation) |
| 2 | MSc | Delft University | 7.5 | - | 8.5 | - | Soil temperature measurement to estimate evaporation. |
| 3 | MSc | Delft University | 7 | Enough | 7 | Enough | - |
| 4 | PhD | Delft University | 8 | - | 8 | - | Aerial photos, LAI (although it is difficult) |
| 5 | PhD | Delft University | 8 | - | 8 | - | Try to measure discharge, modelling the impact of soil moisture & transpiration |
| 6 | Prof | UNESCO-IHE | 8 | - | 8 | - | Discharge measurements (notches) to close the detail water balance, micro station especially evaporation |
| 7 | PhD | UNESCO-IHE | 7 | - | 7.5 | - | Plant's physiological effects like water stress. |
| 8 | MSc | Delft University | 7 | - | 8 | - | A higher resolution is usually better. |
| 9 | MSc | Eindhoven-Deltares | 8 | - | 8.5 | - | - |
| 10 | PhD | Delft University | 7.5 | - | 8 | Try to see sub pixel | Pixel variability (to minimize the interval of images). Sensors for spectrometer through cable trolley/station and aerial photos (for more detail images) |



Table C.3. Indonesian case; the experts' opinions

| No | Title | Institution | Scenario 2 | | Scenario 3 | | Other suggestions |
| | | | Grade | Remarks | Grade | Remarks | Remarks |
|---|---|---|---|---|---|---|---|
| 1 | PhD | Utrecht University | 7 | Absolut value is enough | 7 | - | - |
| 2 | MSc | Delft University | 7.5 | Reach a statistical information of regime | 7.5 | Reach a statistical information of regime | - |
| 3 | MSc | Delft University | 7 | - | 7 | - | - |
| 4 | PhD | Delft University | 8 | With more data we can get new or not increased understanding | 8 | Spatial information is important | Integrate geology and other point measurements at other rivers |
| 5 | PhD | Delft University | 7 | Hydrology engineering | 8 | Measure at more various areas. Start by investigating maps of the different factors; geological, topographical, vegetation, and boundary condition. | More variability means more recommendation to result in a catchment classification with certain discharges, but maybe diverse catchments are depended only on landscape and geology |
| 6 | Prof | UNESCO-IHE | 7 | - | 8 | - | Higher resolution of DEM. |
| 7 | PhD | UNESCO-IHE | 7.5 | - | 8 | - | Maps or information on internet: meteorological data (rainfall & temperature) DEM, soil, geology, & land use. Multiple regression (Q) |
| 8 | PhD | Delft University | 7 | - | 8 | - | Map of the basin. Field survey on all potential places based on the distance from the village etc. Socio-economic studies to answer where to build a MHPP. |
| 9 | MSc | Eindhoven-Deltares | 7 | - | 8 | - | Geology and land use map. |
| 10 | PhD | Delft University | 7 | - | 7 | - | Not important in hydrological science, but just as hydrological measurement. If one goes for uncertainty, then a long time series is needed. A flume (which will be costly) is an option to measure the Q. |





Figure 1. The location of the trenched area, rain gauge, and constructed wells. The study area
is the shaded area on the lower map. Source: Local produced map and Google Earth.

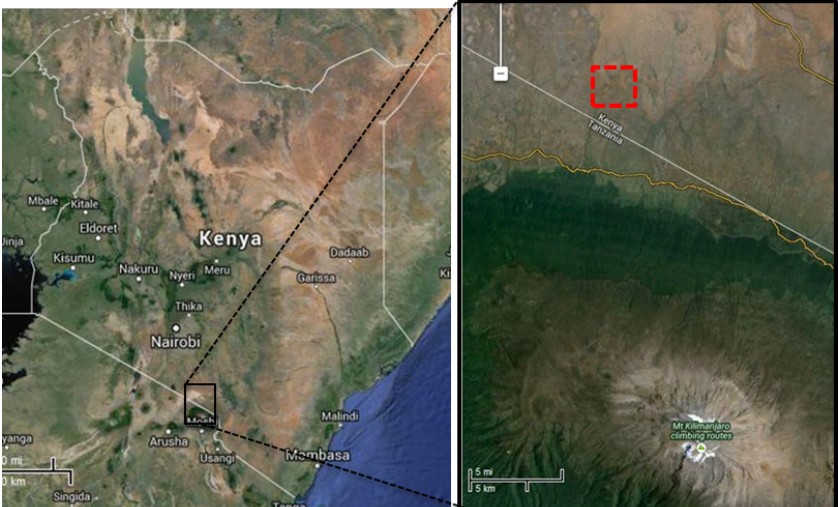

Figure 2. Location of studied contour trenching in Amboseli, Kenya (red dashed line). Source:
Google Maps.



2



5      Figure 3. The result of merged and processed DEM tiles on Maluku islands.





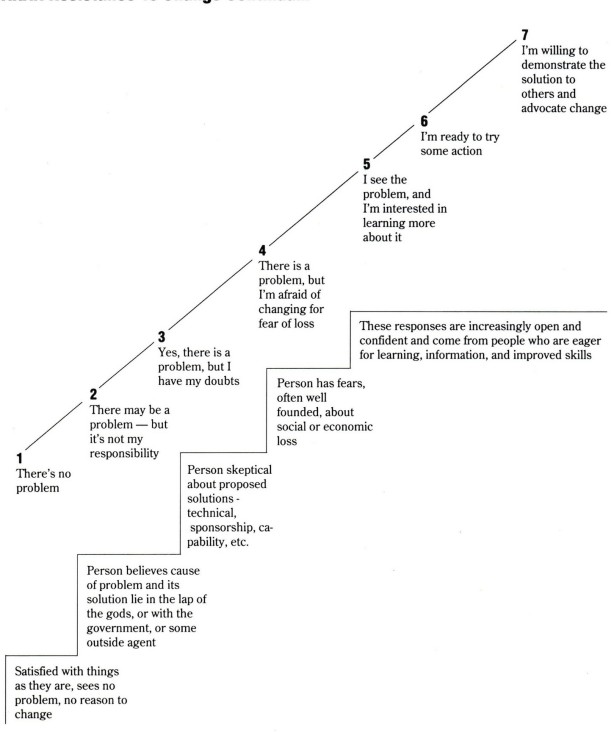

2    Figure 4. The scale of community participation. Source: Srinivasan, 1990, page 162.
3





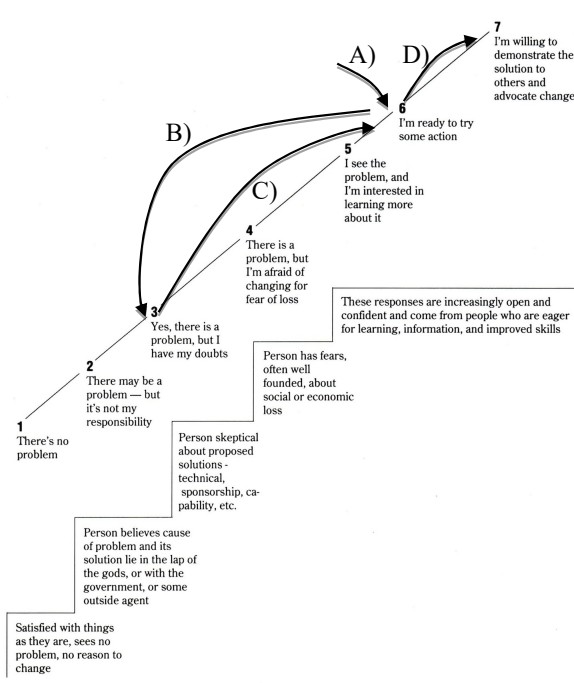

2   Figure 5. The implemented intervention based on the scale of community participation of
3   Srinivasan (1990).





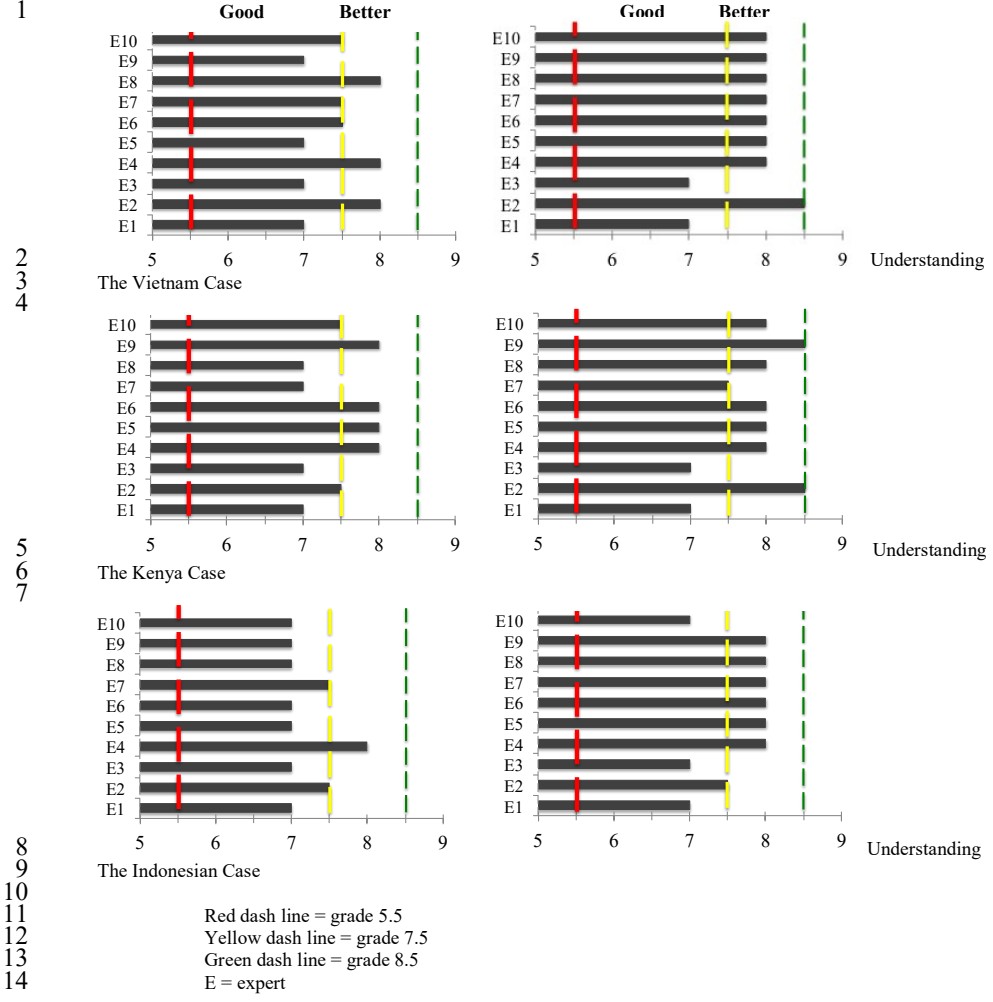

Red dash line = grade 5.5
Yellow dash line = grade 7.5
Green dash line = grade 8.5
E = expert

Figure 6. Summary of three cases; on the left: Scenario 2, right: Scenario 3






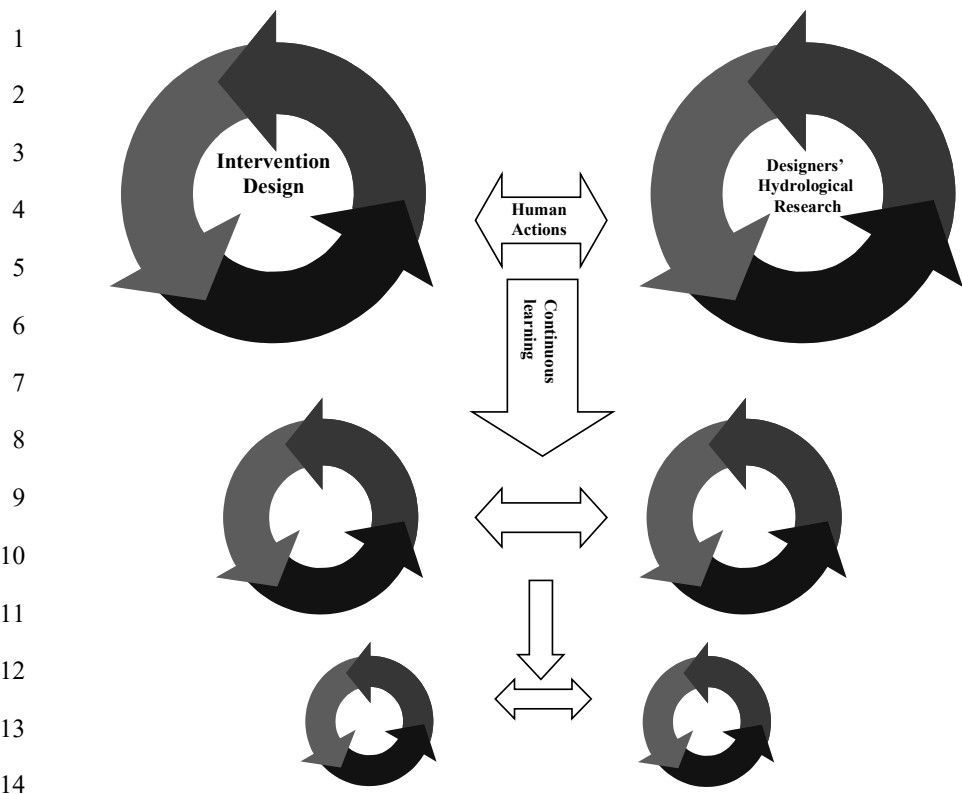

Figure 7. Designing hydrological field research in small-scale intervention (modified from
Ertsen (2002) and Scheer (1996))