# Peer review of "Towards systematic planning of small-scale hydrological"

_Hydrology and Earth System Sciences, 2016_

## Referee Comment (RC1) · Anonymous Referee #1 · 3 Jun 2016

Based on three specific case studies, the paper discussed some of the contingencies and constraints involved when implementing simple hydrologic engineering designs in the context of rural communities in developing countries. The paper raises interesting issues on the practical challenges involved in the implementation of hydrological research, which are rarely discussed in the mainline peer reviewed literature.

The authors have addressed the concerns raised in the discussion on the previous submission:Arguments are now well structured, (for the most part) read easily and are well grounded in the literature. In my opinion, the manuscript is acceptable to be published in HESS, provided the following minor comments are addressed:

p8.l.17-18. The sentence appears self-contradictory. Please reformulate.

[Figure]

p8. l.16-21. It would improve on clarity to give an explicit response to the the research questions of the Kenya case here.

p.8,l.24-p9,l19: Please provide more details on the measurement campaign. Frequency and duration of observations? How do you "extrapolate" discharge to the whole region, based on what data and assumption? I realize this is perhaps not directly relevant to the main point of the paper, but this short description of the project is rather incomplete as is and does not allow to asses the relevance of the study. A bit more detail, perhaps to the level of case study 1, would be really useful, in particularly given that the characteristics of the measurement campaign are precisely what the expert are requested to evaluate in the interview. (p.18, l 15.)

p11 l21: please describe what [#6] means: perhaps refer to a table if appropriate.

p12 l4 and l17: the referred tables do not exist.

p12 l24: please define the acronym.

Section 5: What's the purpose/added value of 5.3 vs 5.4: perhaps provide an introductory paragraph to 5.3. I can see how 5.4 is important, but 5.3 appears perhaps less relevant and (unless you explicitely state the point you're trying to make) should go in appendix.

Section 5: Although the manuscript is generally understandable and pleasant to read, the quality of writing decreases substantially in section 5. The writing style in sections 5.1 and 5.2 is poor and requires more editing in my opinion, particularly at the following locations, where it impedes understanding:

p14.l18-21

p15.l3-7

p16. l19-21

p17 l13-14

p17 l16 (perhaps "conversely" is the right word?)

p17 l21

p30: Tables 2-4 are cluttered and I fail to see how they illustrate or reinforce and point main in the paper (they are refered to only once and in bulk on p14.l1) I would put them in appendix.

p43: Figure 3 is not refered to in the text

p45: Figures 4 and 5 are redundant: please consolidate

p46 Figure 6 is complex, its take away is not immediately obvious and appropriately described in the text. Please (at least) be more explicit about the figure's main point in the caption.

---

## Referee Comment (RC2) · Anonymous Referee #2 · 29 Jun 2016

Dear authors,

Thank you for your laudable attempt to present learnings from field research relating to small-scale water interventions in diverse settings to a broad audience. I am supportive of two things that this paper attempts to do: (i) Provide insights into how managing or failing to manage relationships and participation of local stakeholders can disrupt or improve the success of research projects, (ii) Provide information about the performance of specific small-scale water interventions.

That said, the paper itself is, to be blunt, a mess.

It also raised some significant concerns for me (in particular, the major discussions of

human agency in the paper focus on the potential for people to disrupt installations of instruments/sensors, which is a highly negative, narrow and frankly inappropriate way to discuss the nuanced issue of how researchers should engage with communities). I could not support publication of a document that portrays such a narrow view on this important topic. I would like to refer the authors to Srinivasan et al 2015 (HESS) and Thompson et al 2013 (HESS) - in the first case for an illustration of how research can be designed in response to the concerns and perceptions of a wide range of human stakeholders and agents; and in the second case to review the notion of "use-inspired science" - which again highlights the importance of working with stakeholders when conceiving research. The notion that as researchers we *always* have a degree of choice about where to invest our energies and even how to frame our questions, and that this choice can and should be informed by the needs of local communities seems highly pertinent to the premise of this paper (namely that research should be designed with stakeholders and local human agents in mind) - but the motivations for the research questions being asked is not broached at all.

Leaving aside these concerns about ethics, tone and appropriateness; the paper attempts to do too much - to review the ideas of human agency, to present hydrological results from three case studies, to present a set of frameworks for evaluating social perceptions / willingness to participate in research - apparently setting up a de novo participation scale (?), interviewing experts and generating hypothetical research budgets for alternative conceptions of the completed projects, also with reference to ideas from the RAND corporation. This morass of stuff renders the larger picture of the paper incoherent.

There are ideas here that are worth more exploration. Drawing from experiences documented by social scientists and by groups like RAND Corp may be very useful for researchers. Cautionary tales of how research can be badly derailed if the social context of the research is not managed well are useful. And given the dearth of published research about the effectiveness of many small scale water interventions, the findings

of the research projects should certainly be published (if they have not been already)? But I think it would be valuable to separate studies about hydrology from studies about doing hydrology, so that the paper could really focus on either physical processes or on the intersection of teams of scientists and local communities in a research context.

If the authors are really committed to the importance of presenting their experiences through the lens of community participation / participatory research / studies of science – then there needs to be much more effort put into reading the pertinent literature (the issue of community willingness to support scientific studies is very broad, and extends well beyond the focus of water interventions - yet the issues raised in very different contexts are still pertinent to hydrologic research. Scientists have often messed up community engagement, in ways that will probably look familiar to the authors, and with consequences that will probably seem equally familiar. The authors need to read this literature and take its lessons on board, and incorporate them into their suggested planning frameworks). A paper that focuses on this aspects of their work need not attempt to present the hydrological results of their work which are a distraction. I would suggest that the authors attempt to find well established frameworks through which to classify the kinds of human intervention that occurred, rather than generating their own, and that these frameworks be introduced early and used consistently through the paper.

I'm unclear on the value of the expert surveys and budgeting. If one's only concern in conducting research on small-scale interventions is unit of knowledge / dollar spent, then some of this information could be useful. But bringing everything back of a dollar bottom line seems to potentially repeat some of the mistakes that might have been problematic in the research to begin with. How to put value on the relationships and willingness of communities to e.g. sustain their infrastructure, or work with the NGO again, or to say nice things about the NGO or the intervention to neighboring villages? Should scientists be thinking about these things to? Is there benefit to having boots on the ground rather than relying on satellite observations? How is a scientist a part of and

an important part of a development team? I am worried, that despite the attempts of these authors to take off the blinkers of a technical research team who focus primarily on getting and analyzing data, that some of these ways of thinking still permeate this paper.

I have attached a PDF with many editorial comments. A final note is that the language used is frequently ambiguous or very hard to interpret, and that the paper is often rather poorly organized. The suggestions on the attached may be helpful, but I think that a bigger picture evaluation of the manuscript's aims, presentation of the state of knowledge, and overall message is needed before publication could be entertained.

Please also note the supplement to this comment:
http://www.hydrol-earth-syst-sci-discuss.net/hess-2016-151/hess-2016-151-RC2-supplement.pdf

[Figure]

**Supplement:**

[revised manuscript text omitted]

---

## Author Comment (AC1) · 9 Jul 2016

In this text, we will respond to both reviewers 1 and 2, whom we thank for sharing their thoughts and concerns with us. We are happy to see that both reviewers agree that the topic we discuss is worthy. Both reviewers have indicated that we need to incorporate further improvements in the text, which we will obviously do, using their valuable and detailed suggestions – including additional useful references and our reasoning below.

Where the reviewers disagree on is whether the argument we develop is sound enough. Reviewer 1 concludes that our arguments are now "well structured" and "well grounded in the literature". Reviewer 2 appears to disagree with that statement completely and suggests that we need to write another paper. It will be no surprise to anyone that we

tend to agree more with reviewer 1 on this topic. We do have some good reasons to do, as we discuss below.

Reviewer 2 starts with claiming that we focus "on the potential for people to disrupt installations of instruments/sensors" and considers this as "highly negative, narrow and frankly inappropriate". We do indeed mention disruptions, but we think we are doing so in a slightly more sophisticated way than simple disruption. Actually, we do criticize the narrow view of "theft and vandalism" on page 4 and mention curiosity or disagreement there as well. Furthermore, our Vietnam case clearly shows that we understand why stakeholders decide to join certain interventions or not, by discussing the different motivations and clear actions of stakeholders, and the way our project dealt with that.

We think the Vietnam case is actually a very good example of a project that in close cooperation with the local stakeholders managed the complexity of intervention-based research. Finally, theft and/or vandalism are difficult terms to use as long as motivations are not known – and we do not use them as such for that reason – but we did experience in Vietnam and Kenya disappearance of measuring devices. So, we disagree with the reviewer that we are negative and inappropriate about human interventions.

Reviewer 2 appears to have different expectations about our paper than we have. We did not intend to write how research "can be designed in response to the concerns and perceptions of a wide range of human stakeholders and agents" nor to discuss the "notion of use-inspired science". This is not because we disagree with the idea of reviewer 2 that "needs of local communities" are important, but we did not take that as central topic. Others have done so, as indicated by the reviewer and by our own references as well. We do take up the issue of designing hydrological research "with stakeholders and local human agents in mind", but in another way than reviewer 2 seems to desire.

Based on our statement that human intervention usually results in lower data availability, we conclude that hydrological research should be designed with stakeholders and local human agents in mind, but we do so with a focus on the hydrological research itself. We would agree with reviewer 2 that is a more narrow focus than the reviewer deals with, but that does not make our focus irrelevant. We discuss how hydrological researchers, who usually do not have a background in participatory theories and are responsible for useful and relevant hydrological data gathering, can improve their studies by preparing in a different way. We would argue that even in projects with stakeholder involvement as the reviewer would like to see, surprise will be a part of the hydrological work – as we have seen indeed in Vietnam.

In order to make this argument, we do think we need our material on the three hydrological studies, the details of the different responses of stakeholders, and the focus on costs and benefits in terms of data. We do not focus on those costs and benefits because that would be the only valuable focus in projects in general, but simply because in the type of projects we discuss, research budgets are limited, but everyone does still expect useful hydrological results.

We would be the last one to suggest that our paper offers the final word on the topic, but we maintain that our focus on how hydrologists should plan ahead for surprise and can do so by using a cost-benefit approach is useful, that we did discuss these and other issues in an appropriate way, and that our agreement with reviewer 1 makes sense. We will obviously use the comments of both reviewers to improve our paper in case we are allowed to do so, and include some of our text above to make our reasoning appear even stronger.

---

## Author Response (AR1)

Dear Editor and Referee# 2,

We would like to thank the valuable suggestions of the Editor and Referee #2. Based on the comments that were detailed by Referee #2 and summarized by the Editor, we have revised the paper substantially and answered the Editor's concern as follows:

**(i)     Structure and clarity**
We changed the structure of the paper with more general methodological explanation before the case, even though we keep much of the detail in the sections after the case, as we think in this way we can better explain and illustrate the approach we developed. We also clarified the participation discussion and our position (see ii). Lastly, we clarified the paper's theme in relation to participation (for further answer to iii).

**(ii)    Novel scales for rating participation**
We kept our previous scale for rating participation, since the rating itself is not our focus. Our paper does not intend to create a novel participation scale, but use an existing one to illustrate that stakeholder involvement/participation can be studied and needs to be understood, where surprises should be taken into account. Whatever (novel) scale is presented, we argue that local community participation in hydrological research do have certain motivation(s). However, when it comes to surprises, we will always have to deal (properly; being aware, research plan improvisation, etc.) with them.

**(iii)   Section 2 "Human Agency and Hydrologic Research"**
We have included much more in terms of literature review relevant within our context – small hydrological research versus intervention - in the introduction. Next, we changed the above title to "Human actions as surprises in intervention development and hydrological research". This section describes what to do with surprises – when shifting location of intervention occurs and measuring devices disappear – during the development of intervention and its associated hydrological research. We propose RAND for practical concern and budget scenarios for limited budget issue as the approach in hydrologic field research in small-scale intervention towards human actions. The application of scenarios, specifically on the Vietnam case, is presented in section 5 (Vietnam case).